# Crosstalk between *MIR-96* and *IRS/PI3K/AKT/VEGF* cascade in hRPE cells; A potential target for preventing diabetic retinopathy

**Zeynab Hosseinpoor**[1], **Zahra-Soheila Soheili**[1]*, **Maliheh Davari**[1], **Hamid Latifi-Navid**[1,2,3], **Shahram Samiee**[4], **Dorsa Samiee**[5]

1 Department of Molecular Medicine, National Institute of Genetic Engineering and Biotechnology (NIGEB), Tehran, Iran, 2 Electrophysiology Research Center, Neuroscience Institute, Tehran University of Medical Sciences, Tehran, Iran, 3 School of Biological Sciences, Institute for Research in Fundamental Sciences (IPM), Tehran, Iran, 4 Blood Transfusion Research Center High Institute for Research and Education in Transfusion Medicine, Tehran, Iran, 5 Department of Computer Science, Royal Holloway University of London, Egham, Surrey, United Kingdom

* soheili@nigeb.ac.ir

**Data Availability Statement:** All relevant data are within the manuscript.

## Abstract

Regulation of visual system function demands precise gene regulation. Dysregulation of miRNAs, as key regulators of gene expression in retinal cells, contributes to different eye disorders such as diabetic retinopathy (DR), macular edema, and glaucoma. *MIR-96*, a member of the *MIR-183* cluster family, is widely expressed in the retina, and its alteration is associated with neovascular eye diseases. *MIR-96* regulates protein cascades in inflammatory and insulin signaling pathways, but further investigation is required to understand its potential effects on related genes. For this purpose, we identified a series of key target genes for *MIR-96* based on gene and protein interaction networks and utilized text-mining resources. To examine the *MIR-96* impact on candidate gene expression, we overexpressed *MIR-96* via adeno-associated virus (AAV)-based plasmids in human retinal pigment epithelial (RPE) cells. Based on Real-Time PCR results, the relative expression of the selected genes responded differently to overexpressed *MIR-96*. While the expression levels of *IRS2*, *FOXO1*, and *ERK2* (*MAPK1*) were significantly decreased, the *SERPINF1* gene exhibited high expression simultaneously. pAAV-delivered *MIR-96* had no adverse effect on the viability of human RPE cells. The data showed that changes in insulin receptor substrate-2 (IRS2) expression play a role in disrupted retinal insulin signaling and contribute to the development of diabetic complications. Considered collectively, our findings suggest that altered *MIR-96* and its impact on *IRS/PI3K/AKT/VEGF* axis regulation contribute to DR progression. Therefore, further investigation of the *IRS/PI3K/AKT/VEGF* axis is recommended as a potential target for DR treatment.

## Introduction

microRNAs (miRNAs) are a class of small non-coding RNAs that influence gene expression patterns of cellular pathways and regulate diverse biological processes in this way. Their

**Funding:** The National Institute of Genetic Engineering and Biotechnology has allocated Grant 727 to this research project following proposal evaluation and review. The funders had no role in study design, data collection and analysis, decision to publish, or preparation of the manuscript.

**Competing interests:** The authors have declared that no competing interests exist.

interaction with specific sequences in target mRNAs mediates post-transcriptional gene regulation. Recent studies have revealed that in addition to gene silencing, miRNAs can also activate gene expression through particular pathways [1, 2]. They play a leading role in developmental, functional, and pathological conditions in the retina [3]. The microRNA-183 cluster is a prominent family among the most frequently reported miRNAs in the retina, comprising *MIR-183*, *MIR-96*, and *MIR-182*. During retinal development, members of the *MIR-183* cluster are overexpressed as a single transcript and regulate different target genes involved in cellular pathways [4]. Given their pivotal roles, altered expression of the *MIR-183* cluster has been shown to be linked to a variety of human diseases, including diabetes and diabetic vascular complications [5, 6]. miRNA microarray analysis indicated that the *MIR-96* level increased in the retinas of streptozotocin-induced diabetic rats [7]. However, the *MIR-96* expression level was significantly downregulated in the serum of type 2 diabetes patients compared with normal controls [8]. A few studies have explored the potential involvement of miR-96 in the development and progression of diabetic retinopathy (DR). However, the exact mechanisms by which miR-96 influences the development of DR remain unclear and require further investigation. A recent study has investigated how the target genes regulated by miR-96 are connected to the signaling pathways associated with DR [9]. Diabetic retinopathy, as one of the most common microvascular complications of diabetes, remains a dominant cause of visual impairments in diabetic patients. In the early stages of DR, the hyperglycemic condition contributes to vasodilation and increased retinal metabolism, pericyte apoptosis, microaneurysm, and eventually retinal microvascular damage. Advanced stages of DR, characterized mainly by neovascularization, culminate in severe vision impairments due to vitreous hemorrhage or detachment of the retina [10, 11]. Altered retinal metabolism and elevated levels of inflammatory molecules affect multiple parts of the retina, including the vascular network, neuronal cells, choroid, and retinal pigment epithelium (RPE) [12]. RPE cells secrete several growth factors and cytokines, including vascular endothelial growth factor (*VEGF*), pigment epithelium-derived factor (*PEDF*), monocyte chemoattractant protein-1 (*MCP-1*), interleukin-6 and 8 (*IL-6*, *IL-8*), and matrix metalloproteinase (*MMPs*). Studies have shown that high glucose exposure modifies the RPE proteome and secretome and can cause alterations in cell structure and function [13].

Recent studies have identified signaling pathways involved in DR development and progression. The most known pathways are insulin signaling, *VEGF* signaling, *IL-6* signaling, and *PI3K-AKT* signaling pathways [14]. According to the latest research, *MIR-96*, as well as other regulatory factors is involved in the control of signaling molecules. For instance, upregulation of *MIR-96* leads to insulin signaling impairment through relevant repressing signaling molecules such as insulin receptor (*INSR*) and insulin receptor substrate-1 (*IRS1*) [15, 16]. IRS proteins are cytoplasmic receptors involved in insulin action by activating the phosphoinositide 3-kinase *(PI3K)*/AKT pathway. Studies showed that *IRS2* is implicated in type 2 diabetes, and its expression is upregulated in DR mouse models [17, 18]. A previous study suggested activating the *PI3K/AKT* pathway upregulates *VEGF* expression in vascular cells [18, 19]. Moreover, a new study determined that *MIR-96* acts as a modulator of multiple angiogenic factors, including *VEGF*, *ANG2*, and *VEGFR2*, during pathological conditions in the retina [20]. These suggest that the *IRS/PI3K/AKT* axis could be counted as a possible target in DR prevention and control.

Considering the involvement of the *MIR-96* target genes in specific signaling pathways, the present study was designed to achieve an understanding of *MIR-96*'s roles in molecular alterations in the retina. To that end, *MIR-96* target genes were provided using four miRNA target prediction online databases, including TargetScanHuman 8.0, miRTarBase 9.0, miRDB, and miRWalk 3. The results of three of these databases include predicted miRNA targets using bioinformatics tools. MiRTarBase 9.0 provides miRNA targets with a higher confidence level

validated experimentally by reporter assay, western blot, microarray, and next-generation sequencing experiments. The main signaling pathways linked to *MIR-96* were identified using mirPath. By analyzing the data of the marked target genes in enriched KEGG pathway maps provided by DIANA-mirPath, the top co-expressed genes were determined utilizing GeneMANIA. In addition, the functional association network was acquired from the STRING online database. Taken together, the results provided a ranked list of genes to investigate the effects of *MIR-96* overexpression in human RPE cells.

## Materials and methods

### In silico target research

To establish a comprehensive understanding of the target interactions for *MIR-96*, the in silico target research was conducted using multiple prediction tools and databases. In silico target prediction for *MIR-96* was done using TargetScan 8.0 (http://www.targetscan.org/vert_80/) [21], miRDB (http://mirdb.org/) [22], miRTarBase 9.0 (https://mirtarbase.cuhk.edu.cn/) [23], and miRWalk 3 (http://mirwalk.umm.uni-heidelberg.de/) [24]. Top-scored genes were identified and integrated with literature search and text mining results obtained from the EVEX database (http://evexdb.org/) [25]. Subsequently, the resulting gene list was introduced to STRING 11.5 (https://string-db.org/) [26], and the functional interactions among identified proteins were visualized using the Cytoscape stringApp (version 3.9.1) [27].

The miRNA-pathway interactions were predicted using DIANA mirPath v.3 (https://dianalab.e-ce.uth.gr/html/mirpathv3/index.php?r=mirpath) [28], and the Kyoto Encyclopedia of Genes and Genomes (KEGG) pathway enriched terms were identified through Metascape website (http://metascape.org/) [29]. In addition, the DisGeNET database (https://www.disgenet.org/) [30–32] was explored to retrieve DR-associated genes, and the gene-disease associations (GDA) network was visualized using the DisGeNET Cytoscape app [33]. This database covers a wide range of diseases and can be a valuable resource for studies on genes associated with human diseases. Fig 1 illustrates a schematic overview of the bioinformatic tools conducted in this study.

### Vector constructs

Following the procedure of the Molecular Cloning Laboratory Manual [34], genomic DNA was isolated from hRPE cells. The *MIR-96* sequence was amplified from genomic DNA using polymerase chain reaction (PCR) (primer sequences: 5′- CGGGGTACCACCGAAGGGCCATA AACAGA - 3′ (Forward), and 5′- CTAGCTCGAGAGTGTAAGGCGATCTGGCT - 3′ (Reverse)). Extra base pairs were added to 5′ end of the primers to increase the cleavage efficiency of restriction enzymes. The purified PCR product was digested with XhoI and KpnI restriction enzymes and inserted into the pAAV-MCS (a human AAV-2 vector included in the AAV Helper-Free System designed by ©Agilent Technologies, USA) that had been previously manipulated in our lab to incorporate the enhanced green fluorescent protein (*eGFP*)-intron cassette [35]. Thereafter, *Escherichia coli* XL10 bacteria (Agilent, USA) was transformed by the ligation product by the heat shock method. Bacterial clones comprising *MIR-96* were identified using plasmid mini-preparation and digestion. Subsequently, the sequences of *MIR-96* and *eGFP*-intron fragments were verified using DNA sequencing analysis (Bioneer Corporation, South Korea). The pAAV-MCS-*eGFP*-int was also employed as a control vector.

### Human RPE cell culture and transfection

The human RPE (hRPE) cell line used in the present study was derived from the previously isolated, characterized, and established HRPE-2S cell line [36]. The cells were cultured in

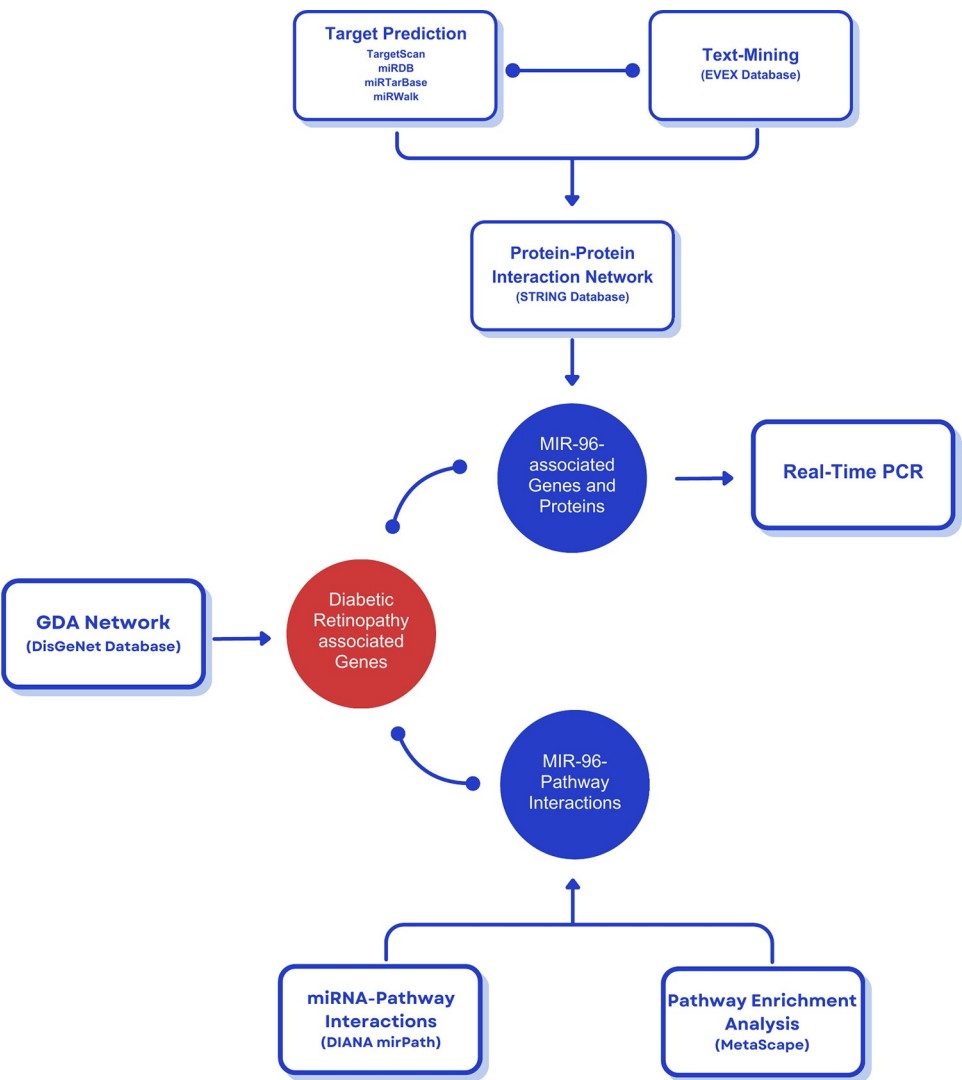

**Fig 1. A schematic overview of the bioinformatic tools conducted in this study.** GDA = Gene-Disease Associations.

DMEM/F-12 mixture (Gibco, Thermo Fisher Scientific, USA) supplemented with 10% fetal bovine serum (FBS, Biowest, France), penicillin (64 μg/ml), and streptomycin (100 μg/ml) in T25 culture flasks (SPL Life Science, Korea). Cells were incubated in a humidified 37 ˚C incubator with 5% $CO_2$ (Binder, USA). For transfections, hRPE cells were seeded in a six-well plate (SPL Life Science, Korea). After 24 hours (60–70% confluency), the cells were transfected with the *MIR-96* expression vector by the calcium phosphate-mediated transfection method. For each well, 5 μg of the desired plasmid was added to sterile dd$H_2O$ and $CaCl_2$ 2M. The $CaCl_2$/DNA mix was added dropwise to an equal amount of HBS 2X (HEPES Buffered Saline) while vortexing. Twenty minutes post-incubation at room temperature, the transfection mixture, containing calcium phosphate-DNA particles was added dropwise and evenly over the cell culture medium. After 6 hours, the medium was replaced with fresh DMEM/F-12 + 10% FBS, and the cells were harvested 72 hours later for further experiments.

## RNA extraction and RT-qPCR analysis

Total RNA was isolated from both transfected and control hRPE cell cultures (which were transfected by pAAV-*MIR-96-eGFP*-int or pAAV-MCS-*eGFP*-int cassette as control) using TriPure Isolation Reagent (Roche, Germany). Mature *MIR-96* expression was quantified following the stem-loop reverse transcription-quantitative polymerase chain reaction (RT-qPCR) procedure [37]. Initially, RNA samples were reversely transcribed by a specific stem-loop primer using the Moloney murine leukemia virus (MMLV) enzyme (Qiagen Inc., USA), then the synthesized complementary DNA (cDNA) was used as a template to amplify *MIR-96* with a specific forward primer and a universal reverse primer. The Real-Time PCR program was as follows: 15 min at 95 ˚C for activation, 15 s at 95 ˚C, and 60 s at 60 ˚C for 45 cycles. The expression level of *MIR-96* transcripts was normalized to the expression level of *RNU48* as endogenous control. To quantify gene expression in transfected hRPE cells, RNA samples were reversely transcribed to cDNAs, using a mixture of oligo (dT) and random hexamer primers and reverse transcriptase (Qiagen Inc., USA). Real-time PCR was performed with QuantiFast SYBR® Green PCR Kit (Qiagen, Germany, Catalog no. 204143) master mix and specific primer pairs of the genes, using Taq DNA polymerase (Qiagen Inc., USA). The utilized Real-Time PCR program was as follows: 10 min at 95 ˚C for activation, 15 s at 95 ˚C, and 60 s at 60 ˚C for 45 cycles. The expression level of genes was normalized to the expression level of glyceraldehyde 3-phosphate dehydrogenase (*GAPDH*) as a verified housekeeping gene. The $2^{-\Delta\Delta CT}$ method was used to calculate and analyze relative changes in gene expression for both the control and *MIR-96*-transfected groups. The relative expression of each gene in the transfected group was reported compared to the control group. The results are representative of at least three independent experiments with three replicates.

## Cell viability assay (MTT)

Cellular metabolic activity, an indicator of cell viability, proliferation, and cytotoxicity was assayed in transfected hRPE cells (pAAV-*MIR-96-eGFP*-int, and control pAAV-MCS-*eGFP*-int transfected hRPE cells) compared to non-transfected control. Both transfected and non-transfected cells were seeded at a density of $1 \times 10^5$ cells per well in 96-well plates (SPL Life Science, Korea). Following overnight incubation at 37 ˚C, the cells were treated with 10 μl 3-(4,5-dimethylthiazol-2-yl)-2,5-diphenyl-2H-tetrazolium bromide (MTT) (5 mg/ml, Sigma-Aldrich, USA). After 4 hours, the medium containing MTT was gently removed, and 100 μl dimethyl sulfoxide (DMSO) (Sigma-Aldrich, USA) was added to each well to dissolve formazan crystals. Thereafter, cell viability was determined by measuring optical density at 580 nm using a microplate reader. Non-transfected hRPE cells were considered for normalizing absorbance data.

## Statistical analysis

All transfections were independently done at least three times. RT-qPCR was performed in three independent experiments with three replicates in each experiment. Significant statistical differences were evaluated by one-way (to compare the relative expression levels among the candidate genes) and two-way (to assess the changes in gene expression at different time points) analysis of variance (ANOVA). Statistical analyses were carried out using GraphPad Prism version 8.0.1 (GraphPad Software, San Diego, California USA, www.graphpad.com). The data were reported as the mean ± standard error of the mean, and P values less than 0.05 were considered statistically significant.

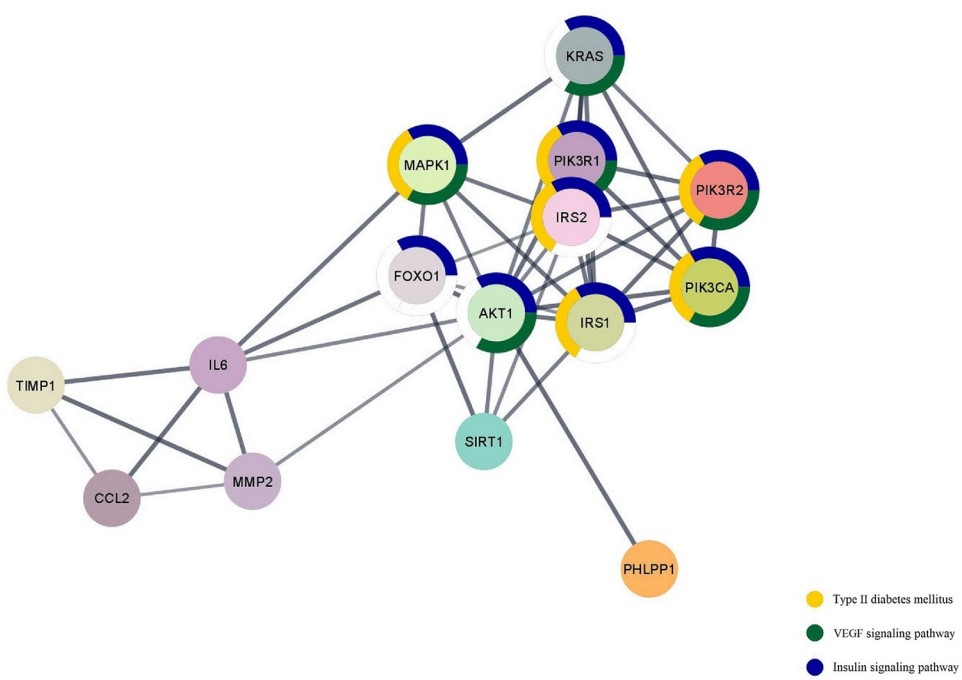

**Fig 2. KEGG pathway enrichment and PPI network of *MIR-96* associated genes.** The thickness of the connecting lines indicates the confidence level. Node borders with different colors present the specific biological pathway. KEGG = Kyoto Encyclopedia of Genes and Genomes; PPI = Protein-Protein Interaction; *VEGF* = Vascular Endothelial Growth Factor.

## Results

### Bioinformatics analysis

We determined overlapping target genes from the four miRNA target prediction databases. The resulting genes were then integrated with text mining outcomes, extended, and visualized using stringApp (Fig 2). The functional enrichment analysis also showed that the target genes are mainly enriched in pathways including type II diabetes mellitus, *VEGF* signaling pathway, and insulin signaling pathway (Table 1), indicating that these genes could be related to several biological processes underlying diabetic retinopathy.

DIANA-mirPath analysis using the Tarbase algorithm with a threshold value of $P < 0.05$, and false discovery rate (FDR) correction implied that *MIR-96* could be involved in the regulation of insulin signaling pathway. Metascape extracted all the protein-protein interactions among the input genes from the PPI data source and formed a PPI network. To identify densely connected proteins, the molecular complex detection (MCODE) algorithm was then applied to this network. Gene ontology (GO) enrichment analysis showed that the MCODE1 network components were significantly enriched in the insulin receptor signaling pathway and type II diabetes mellitus (Fig 3).

**Table 1. Summary of KEGG enrichment analysis.**

| Pathway | Description | Strength | False Discovery Rate |
|---|---|---|---|
| hsa04930 | Type II diabetes mellitus | 2.23 | 2.05e-11 |
| hsa04370 | *VEGF* signaling pathway | 2.14 | 4.92e-11 |
| hsa04910 | Insulin signaling pathway | 1.95 | 1.39e-14 |

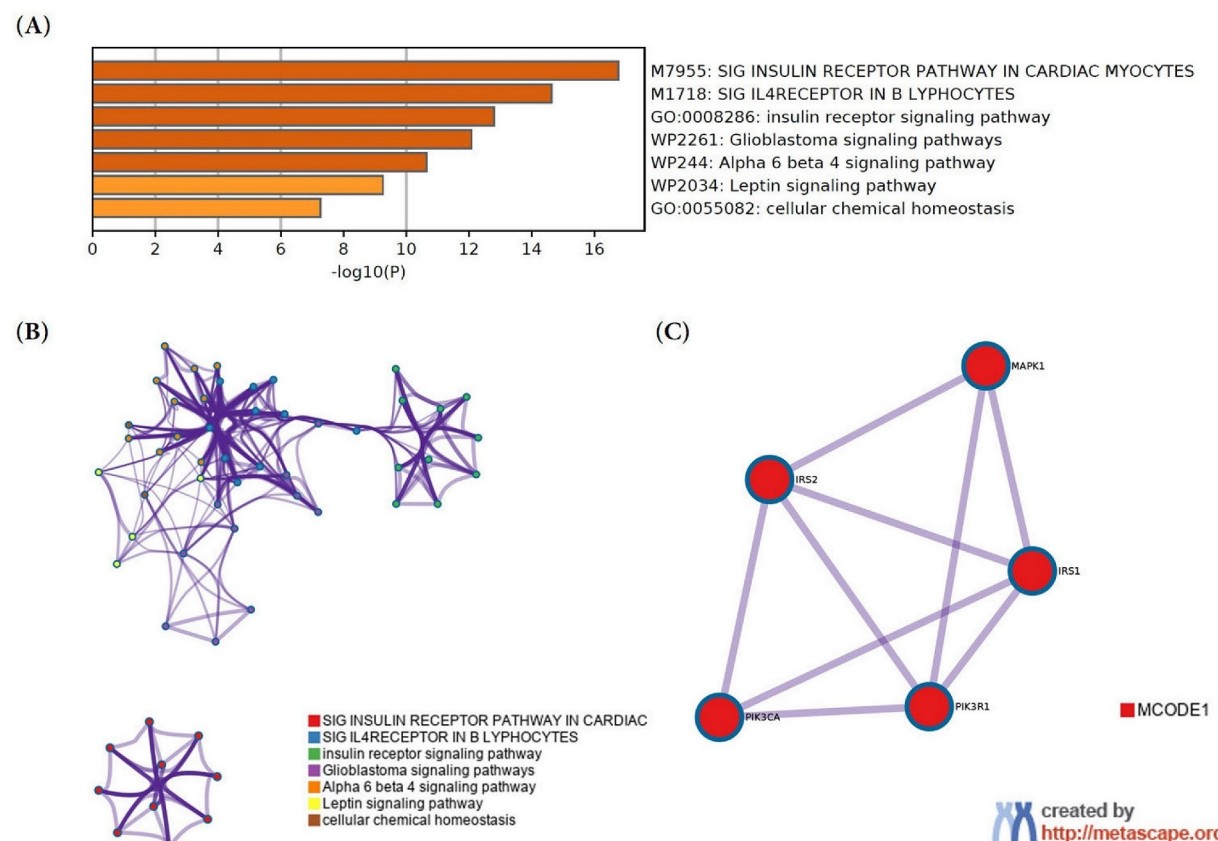

**Fig 3. Gene ontology enrichment analysis.** (A) GO and KEGG pathway enriched terms recognized using Metascape website, sorted by -log10. (B) Network layout of representative terms visualized by Cytoscape. Each term was represented by a circle node. The colors describe the clusters' identities. The thickness of the connecting line represents the similarity score. (C) The most significant MCODE model. GO enrichment analysis was applied to MCODE1, and the term "type II diabetes mellitus" was retained from the top three best p-value terms. GO = Gene ontology; KEGG = Kyoto Encyclopedia of Genes and Genomes; MCODE = Molecular Complex Detection.

To explore related genes that are likely involved in the molecular mechanisms underlying diabetic retinopathy, DR-associated genes (altered expression) were retrieved from the DisGe-NET database (v7.0) and gene-disease associations (GDAs) were visualized by the DisGeNET Cytoscape app (Fig 4).

## Construction of the *MIR-96* expressing vector, transfection and analysis of *GFP* expression

*MIR-96* gene was amplified from human genomic DNA. The final PCR product (424 bp) was inserted into a pAAV-MCS-*eGFP*-int plasmid (5506 bp, control vector) to produce a 5.9-kb recombinant vector (Fig 5A). The 1.3-kb DNA fragment containing *MIR-96*, *eGFP*-int, and cloning junction sequences was verified by DNA sequencing. Next, the recombinant plasmids were transfected to hRPE cells. 48 hours post-transfection, the cells were inspected for *eGFP* protein expression using fluorescence microscopy. As illustrated in Fig 5B and 5C, both hRPE cultures transfected with either pAAV-*MIR-96*-eGFP-int or pAAV-MCS-*eGFP*-int display over 90% *eGFP*-expressing cells.

**(A)**

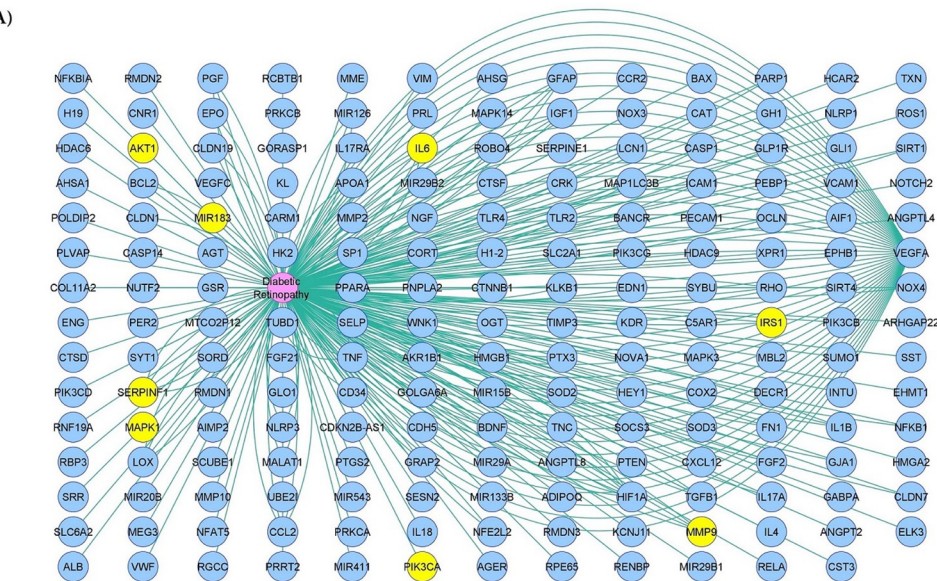

**(B)**

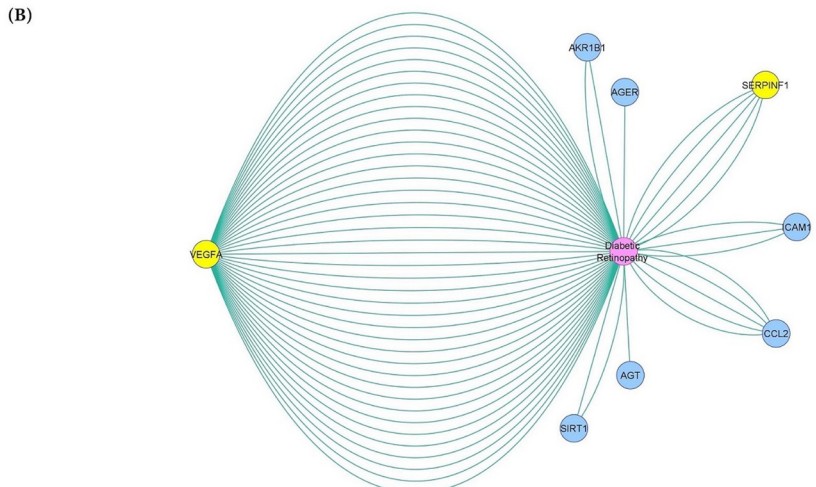

**Fig 4. Network of differentially expressed genes associated with diabetic retinopathy.** (A) Association of common differentially expressed genes with diabetic retinopathy (DR) according to the DisGeNET database. (B) Top scored DR-associated genes. The enriched signaling pathway interrelated genes were highlighted in yellow, and the score ranges from 0.5 to 1.0.

## Overexpression of *MIR-96* in the transfected hRPE cells

The result of stem-loop RT-qPCR, when compared with hRPE cells transfected with pAAV-MCS-*eGFP*-int as control, revealed that *MIR-96* was upregulated by more than 82-fold in the transfected hRPE cells 48 and 72 h post-transfection (Fig 6A).

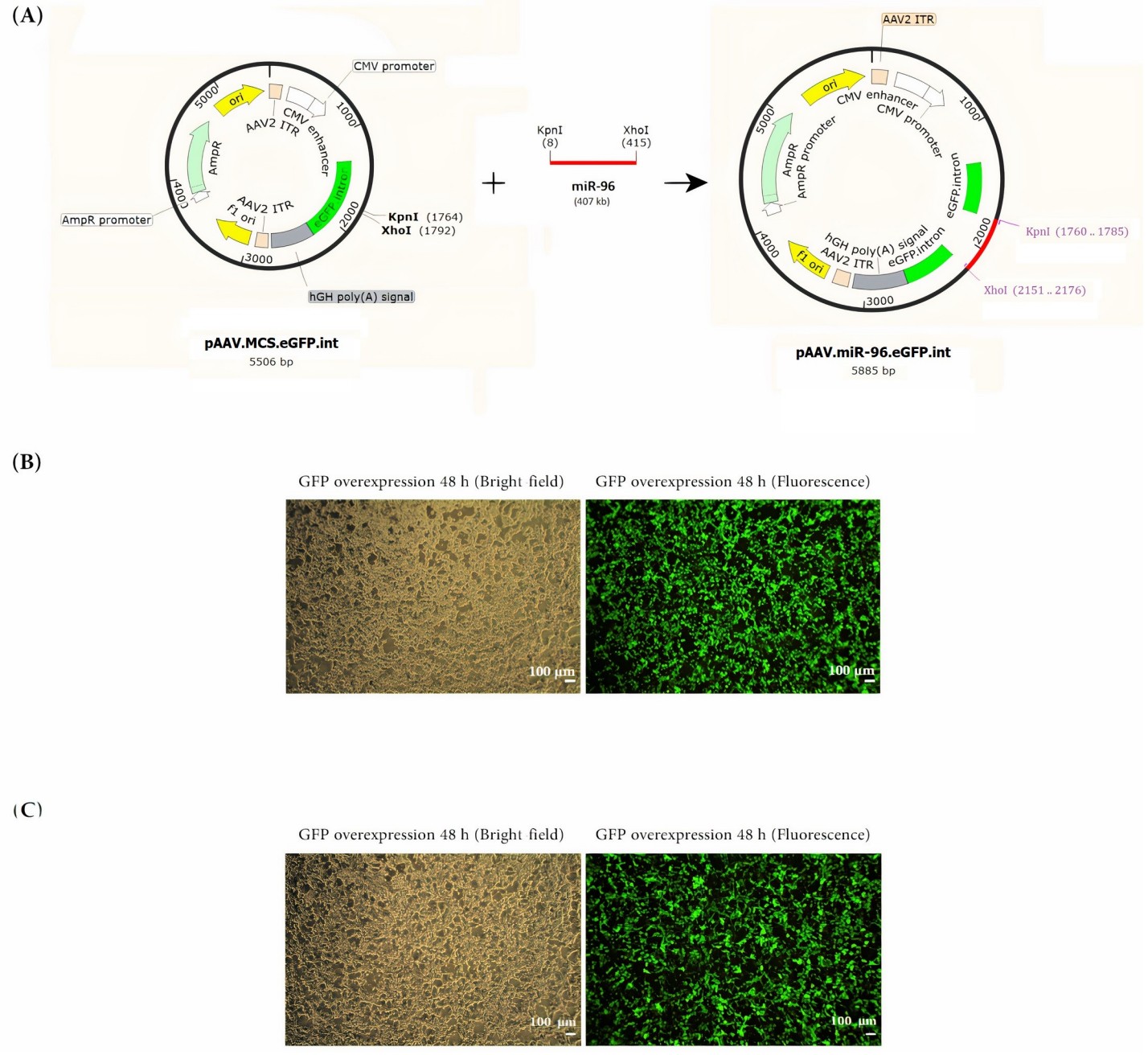

**Fig 5. Schematic representation of pAAV-*MIR-96-eGFP*-int construct and hRPE cells' transfection.** (A) *MIR-96* fragment was inserted into pAAV-MCS-*eGFP*-int cassette (5.5 kb) to construct pAAV-*MIR-96-eGFP*-int vector (5.9 kb). (B) *GFP* expressing hRPE cells, 48 h after transfection of the recombinant vector, and (C) *GFP* expressing hRPE cells, 48 h after transfection of the control vector. The fluorescence microscopy image demonstrates more than 90% of both cultures were *GFP* positive. hRPE = human Retinal Pigment Epithelium; *AmpR* = ampicillin resistance gene; CMV = cytomegalovirus; AAV2 = adeno-associated virus-2; ITR = inverted terminal repeats; *GFP* = green fluorescent protein.

## Impact of pAAV-*MIR-96-eGFP*-int overexpression on hRPE cell viability

We further evaluated the effect of *MIR-96* overexpression on the viability of the transfected hRPE cells in a time-dependent manner. MTT assay revealed that overexpression of *MIR-96*

**(A)**

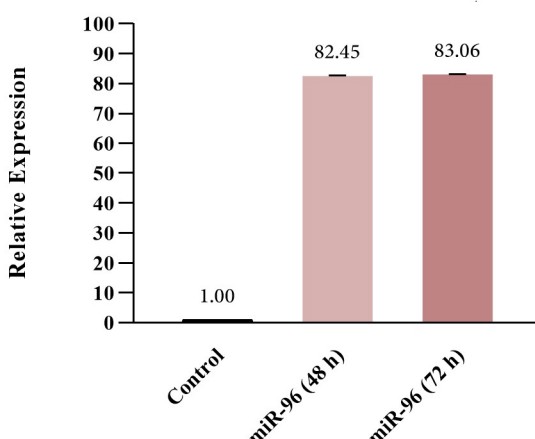

**(B)**

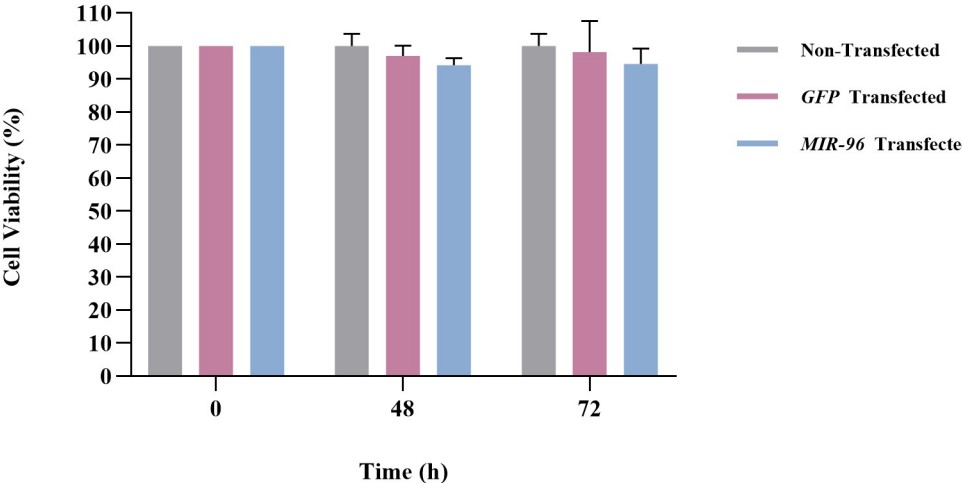

**Fig 6. Effects of *MIR-96* on cell viability.** (A) The relative expression level of *MIR-96* following hRPE cell transfection (at 48 and 72 hours). (B) Cell viability analysis using MTT assay. Data expressed as mean ± standard error of the mean (SEM). $P < 0.05$. MTT = 3-(4,5-dimethylthiazol-2-yl)-2,5-diphenyl-2H-tetrazolium bromide; *GFP* = green fluorescent protein.

had no statistically significant effect on cell viability when compared with the control group ($P = 0.84$, one-way ANOVA, Fig 6B).

## Effect of *MIR-96* overexpression on target genes' expression

To explore the possible impacts of *MIR-96* overexpression on insulin signaling pathway mediators and DR-related genes, the expression level of the nominated genes was assessed using

**(A)**

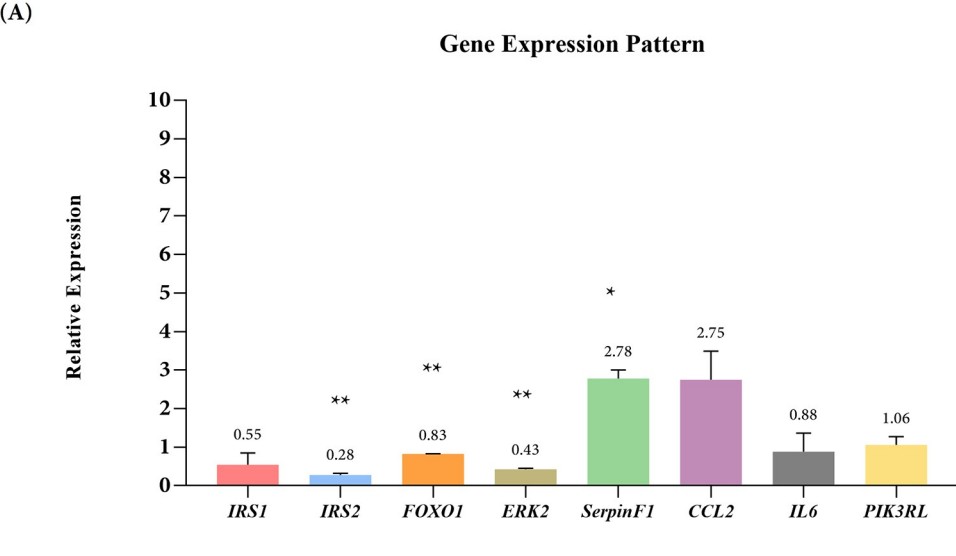

**(B)**

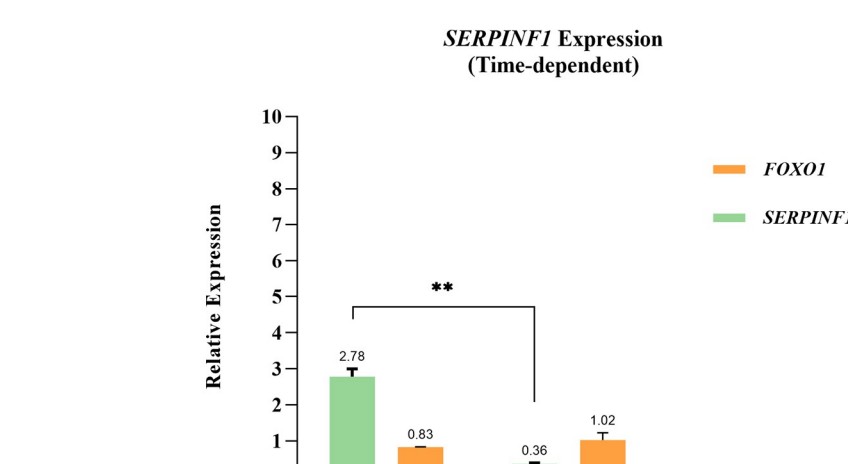

**Fig 7. Relative expression of selected genes in transfected hRPE cells.** Error bars represent means ± standard error of the mean (SEM). *P < 0.05, **P < 0.01. *IRS1* = Insulin Receptor Substrate 1; *IRS2* = Insulin Receptor Substrate 2; *FOXO1* = Forkhead box protein O1; *ERK2* = Extracellular signal-Regulated Kinase 2; *SERPINF1* = Serpin Peptidase Inhibitor, Clade F, Member 1; *CCL2* = Chemokine (C-C motif) Ligand 2; *IL6* = Interleukin 6; *PIK3R1* = Phosphoinositide-3-Kinase Regulatory Subunit 1. The p-values of *IRS2*, *FOXO1*, *ERK2*, and *SERPPINF1* are 0.001, 0.001, 0,002, and 0,014, respectively.

RT-qPCR. As represented in Fig 7, the expression levels of *IRS2*, *FOXO1*, and *ERK2* genes were significantly downregulated (by 0.3, 0.8, and 0.4-fold, respectively) 48 hours post-transfection in pAAV-*MIR-96-eGFP*-int transfected group compared with the control. Furthermore, SERPINF1 which displayed higher expression (2.7- fold) up until 48 h, decreased (0.4-fold) within 72 h after transfection compared with the control group.

## Discussion

Recently, there has been considerable interest in comprehending the diverse molecular mechanisms behind DR to enhance the effectiveness of therapeutic strategies. Anti-*VEGF* therapy is

a reliable and responsive treatment for proliferative diabetic retinopathy (PDR) and can also reduce the severity of DR in non-proliferative diabetic retinopathy (NPDR) patients. Despite its efficacy, reports show that DR lesions often recur quickly and underlying retinal ischemia also remains unchanged. Although anti-*VEGF* therapy is considered the main treatment option for DR, recent studies focus on the different pathways involved in the development of DR in the retina. Identifying the components of these pathways and designing effective drugs can provide new therapeutic options, whether used alone or in combination with anti-*VEGF* drugs, may enhance the effectiveness of treatments. Therefore, we need to target new pathophysiological pathways and stabilize drug effects [38, 39]. Regarding the importance of DR-associated signaling pathways in developing new pharmaceutical agents targeting non-*VEGF*-driven pathways, the current study aimed to identify potent components of the related signaling pathways.

Based on the results of the KEGG pathway enrichment analysis, *MIR-96* target genes are predominantly enriched in signaling pathways activated by insulin (phosphoinositide 3-kinase) and *VEGF*. Integrating the protein-protein interaction (PPI) network of *MIR-96* associated genes with an in vitro study of *MIR-96* overexpression in hRPE cell culture has shown that *IRS2* and *FOXO1* levels were significantly decreased after 48 hours in pAAV-*MIR-96-eGFP*-int transfected hRPE cells.

Insulin signaling requires IRS molecules to regulate glucose metabolism in the cells. Among IRS protein family members, *IRS1*, along with *IRS2* are essential molecules for the insulin signaling pathway. Recent evidence revealed that the *IRS2* level has been elevated in mouse models of DR [18]. Differently, *IRS2* expression in macrophages was downregulated by hyperinsulinemia which is often associated with type 2 diabetes [40]. To regulate *IRS2* expression, *FOXO1* binds to the insulin-responsive elements (IRE) of the IRS2 promoter, upregulating *IRS2* expression and activating the *PI3K/AKT* pathway in the process. There is, however, no clear understanding of how *IRS2* dysregulation occurs [41, 42].

It has also been demonstrated that insulin regulates *VEGF* expression in several cell types, including epithelial cells, and thereby contributes to DR progression. *VEGF* by itself can damage the tight junctions and cause dysfunction in RPE cells [43]. As far as we know, insulin upregulates the expression of *VEGF* mainly through the *PI3K/AKT* axis. Data support the idea that the *IRS/PI3K/AKT/VEGF* axis could be a potential target for the treatment of DR and accordingly, it should be further investigated to find new therapeutic targets [44].

Considering previous research, our findings suggest that the alteration of *MIR-96* and its effects on *IRS/PI3K/AKT/VEGF* axis regulation and therefore the enhancement of *VEGF*, act as a facilitator in DR progression. According to Ji et al., suppression of *IRS2* by *MIR-7a* inhibits *PI3K/AKT* cascade proteins in retinal pericytes and endothelial cells of DR mouse models. They suggested that the upregulation of *MIR-7a* and targeting *IRS2* can inhibit *VEGF* and also the invasion capability of retinal cells [18]. In the current study, we found that the overexpression of *MIR-96*, contrary to other in vitro experiments [18, 45], can eventually result in the downregulation of *PI3K/AKT* cascade proteins, such as *IRS2*. Although the *IRS2* expression differs from previous studies, it can nevertheless be argued that there is probably an interconnected system including epigenetic control, upstream genes, and miRNAs that can play a role in the regulation of *IRS2* expression in diabetic patients [41, 42]. In addition, our studies were performed on non-diabetic retinal cells in which the *MIR-96* was overexpressed. Despite the decline of *IRS2*, the expression of *IRS1* remained unaltered which was in line with the findings from the study reviewed by Pitale et al. [45, 46].

A more recent study by Zolfaghari et al. [9] showed that despite the general similarity of retinal pigment epithelial cells in humans and mice, there are differences that should be considered for developing novel treatments [47]. A comprehensive review of the cellular

characteristics suggested significant differences associated with each species. For example, RPE cell proliferation was reduced in mice as a result of *MIR-96* overexpression, while it remained unchanged in human retinal cells. Nonetheless, the decrease in the expression level of *FOXO1* in human cells concurs well with the previous study on the retina of human donors with DR [9].

*FOXO1*, which activates through the insulin-stimulated *PI3K* pathway and contributes to diabetes hyperglycemia, is a direct target of *MIR-96* [48, 49]. Moreover, *FOXO1* has now been identified as a potential DR-specific diagnostic and therapeutic gene, and its aberrant expression is associated with the pathogenesis of DR. It has been reported that several signaling pathways including *MAPK* regulate the activity of FOXO proteins in response to hyperglycemic conditions in diabetes [50, 51].

Among autophagy-related genes identified by bioinformatics analysis of mRNA chip of DR samples, *ERK2* (*MAPK1*) was found to be downregulated [52]. In the present study, *ERK2* gene expression level showed a significant reduction in pAAV-*MIR-96*-*eGFP*-int transfected hRPE cells. Several studies indicate that *ERK1/2* plays a critical role in the development of DR [53–55]. It has been suggested that in addition to the *PI3K/AKT* signaling pathway, the *ERK1/2* (p44/42 MAP Kinase) pathway may also contribute to long-term *VEGF* upregulation. As shown in previous studies, *ERK1/2* can regulate *VEGF* expression by acting at the *VEGF* promoter. Furthermore, it has been shown that the *ERK* signaling pathway contributes to *VEGF* release in the retinas of diabetic rats [56].

Consistent with RT-qPCR results, Gene-disease associations (GDAs) provided by the DisGeNET (v7.0) database showed that among the DR-related genes, there are genes (such as *IRS1*, *PIK3CA*, *AKT1*, and *MAPK1*) related to the insulin signaling pathway and its downstream pathways whose expression has changed. In addition to these genes, *MIR-183*, according to a recent study, was also upregulated substantially in the retinas of DR rat models. The upregulation of *MIR-183* activated the *PI3K/AKT/VEGF* signaling pathway promoting angiogenesis and endothelial cell proliferation [57].

Several studies have shown that *PEDF* (*SERPINF1*) is highly expressed in RPE cells, which affects retinal vasculature, and sustains retinal function. *PEDF* is a versatile protein that can inhibit the development of DR due to its antioxidant effect [58–61]. It is important to note that the balance between the expression of *VEGF* and *PEDF* in RPE cells has a profound impact on retinal vessels so an increase in *VEGF* and a decrease in *PEDF* can lead to the development of retinal neovascularization [62]. Likewise, studies have linked DR pathogenesis to decreased levels of *PEDF* [63]. In the present study, we observed that *SERPINF1* expression started to decline over time after transfection, despite the elevation in the early hours. As the most potent neovascularization inhibitor in the eye, *SERPINF1* reduction can affect the protection of retinal cells, thus the RPE becomes defective and unable to digest photoreceptor outer segments, allowing the retina to degenerate [64, 65].

It has been shown that *PEDF* signals through both *MAPK/ERK* and *AKT* signaling pathways to regulate gene expression [66]. It can also provide protection by activating the *ERK* pathway. Indeed, *ERK* activation has a protective function in the rescuing of RPE cells in an oxidative environment [67, 68]. In addition, *PEDF* inhibits endothelial cell proliferation by regulating the *MAPK/ERK* pathway [69]. Furthermore, there is evidence that *PEDF* can activate the *PI3K/AKT* pathway to reduce cytotoxicity in RPE cells that are being exposed to oxidative stress [70]. *SERPINF1* is also present in the GDAs network and is one of the top-scored DR-associated genes. Delivering *SERPINF1* in order to repress angiogenesis in the retina revealed that this approach was effective at inhibiting intravitreal neovascularization [71, 72].

Studies have shown that the regulation of the *MMP2* gene is closely associated with the activation of the *PI3K/AKT* pathway [73]. Based on recent research [74], the *PI3KCA/IRS1/2/*

*AKT1/IL6/MMP2* axis shown in the PPI network of *MIR-96* associated genes (Fig 2) contributes to retinal neovascularization by enhancing *MMP2* expression through activation of *PI3K/AKT* signaling.

In summary, this study demonstrated that overexpression of *MIR-96* influences the expression of *IRS2*, *FOXO1*, *ERK2*, and *SERPINF1*, consequently disrupting the *PI3K/AKT* pathway, which may contribute to retinal neovascularization and dysfunction. However, it is believed that further research is required to unravel the mechanism by which the *PI3K/AKT* pathway is dysregulated during DR pathogenesis, along with other associated mechanisms. The present study also confirms that the *IRS/PI3K/AKT* axis plays a prominent role in DR pathogenesis, which can be further investigated as a favorable therapeutic target.

## Acknowledgments

We wish to acknowledge staff of Blood Transfusion Research Center for contribution to this work.

## Author Contributions

**Conceptualization:** Zeynab Hosseinpoor, Zahra-Soheila Soheili.

**Data curation:** Zeynab Hosseinpoor, Zahra-Soheila Soheili.

**Formal analysis:** Zeynab Hosseinpoor.

**Funding acquisition:** Zahra-Soheila Soheili.

**Investigation:** Zeynab Hosseinpoor, Maliheh Davari.

**Methodology:** Zeynab Hosseinpoor, Zahra-Soheila Soheili, Maliheh Davari, Hamid Latifi-Navid, Shahram Samiee.

**Project administration:** Zahra-Soheila Soheili.

**Resources:** Zahra-Soheila Soheili, Shahram Samiee.

**Software:** Zeynab Hosseinpoor, Hamid Latifi-Navid, Dorsa Samiee.

**Supervision:** Zahra-Soheila Soheili, Shahram Samiee.

**Validation:** Zeynab Hosseinpoor, Zahra-Soheila Soheili, Shahram Samiee.

**Writing – original draft:** Zeynab Hosseinpoor.

**Writing – review & editing:** Zeynab Hosseinpoor, Zahra-Soheila Soheili, Maliheh Davari, Hamid Latifi-Navid, Dorsa Samiee.

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
