## [Decision Letter · Decision Letter 0]

2 Jun 2024

PONE-D-23-37569Crosstalk between miR-96 and IRS/PI3K/AKT/VEGF cascade in hRPE cells; A potential target for preventing diabetic retinopathyPLOS ONE

Dear Dr. Soheili,

Thank you for submitting your manuscript to PLOS ONE. After careful consideration, we feel that it has merit but does not fully meet PLOS ONE’s publication criteria as it currently stands. Therefore, we invite you to submit a revised version of the manuscript that addresses the points raised during the review process.

Your manuscript was reviewed by two experts and  we received mixed recommendation . Please address the comments as appropriate. 

We look forward to receiving your revised manuscript.

Kind regards,

Partha Mukhopadhyay, Ph.D.

Section Editor

PLOS ONE

 [This work was supported by the National Institute of Genetic Engineering and Biotechnology (NIGEB) through grant no.727.].  

[We would like to acknowledge the National Institute of Genetic Engineering and Biotechnology (NIGEB) for support to conduct this research.]

 [This work was supported by the National Institute of Genetic Engineering and Biotechnology (NIGEB) through grant no.727.]

6. Please provide a complete Data Availability Statement in the submission form, ensuring you include all necessary access information or a reason for why you are unable to make your data freely accessible. If your research concerns only data provided within your submission, please write "All data are in the manuscript and/or supporting information files" as your Data Availability Statement.

Reviewers' comments:

Reviewer's Responses to Questions

**Comments to the Author**

1. Is the manuscript technically sound, and do the data support the conclusions?

Reviewer #1: Yes

Reviewer #2: Yes

2. Has the statistical analysis been performed appropriately and rigorously? 

Reviewer #1: Yes

Reviewer #2: Yes

3. Have the authors made all data underlying the findings in their manuscript fully available?

Reviewer #1: Yes

Reviewer #2: No

4. Is the manuscript presented in an intelligible fashion and written in standard English?

Reviewer #1: Yes

Reviewer #2: Yes

5. Review Comments to the Author

Reviewer #1: This manuscript is series of genes based on gene and protein interaction networks and utilized text mining resources. To examine the miR-96 impact on candidate genes expression, we overexpressed miR-

96 via adeno-associated virus (AAV)-based plasmids in retinal pigment epithelial (RPE) cells. This manuscript is good in shape well described.

I'm happy to accept this manuscript.

Reviewer #2: The authors started from the premise that microRNA from the family of miR-183 is overexpressed in diabetic retinopathy. They chose miR-96, which is increased in animal models for diabetes to study. The authors then induced miR-96 overexpression in a culture of human retinal pigment epithelium (hRPE) cells through an adenoviral vector. They showed changes in genes related to insulin response and diabetes development, which shows that miR-96 may be important to the development of diabetic retinopathy.

In a wide sense, the paper is well-written, with few grammatical problems (pay attention to some definite articles that don’t seem necessary), yet I recommend it to be double-checked by some specialists. Grammar aside, I found the text hard to follow. It lacks connectivity and a cause/effect sense of your experiments. The bioinformatic segment doesn’t seem to be organically inserted in the text, it seems lost in the context. The in vitro part doesn’t feel like it is leading to the biosystem analysis. As far as I understood, your in silico data would be better presented first, to be later confirmed by the in vitro experiments. This manuscript could use some better storytelling so whoever comes to read your paper might follow it with lesser confusion.

For RT-qPCR you used TaqMan probes from Qiagen, right? I would like you to describe the probes you used, including their catalog number, in the Material and Method section. Regarding your PCR results, it is not clear how you analyzed them. You have basically two groups, control and the one treated with the miR-96 overexpressing AAV. For 2ΔΔCt analysis, you need at least two groups, where one of those would be the control and the other(s) would be times-fold its expression. How did you analyze your data, though? Did you test all those genes for both groups and show only the treated group of each gene in Figure 3A? If so, I don’t have much problem with this way of representing the results, since you keep it very clear in the text, and in the figure legend what you did. For Figure 3B, what was the statistical analysis you used?

Nonetheless, about the statistics, was there any normality test used? It is important to determine if the data was properly analyzed by the tests you described since they are only for normal-distributed data. Table 1 description (line 220) is weird. A P-value equal to or below 0.05 is already statistically significant, for what you described in the methods. As a matter of fact, Table 1 seems redundant. If you want to highlight the p-values, just add them in Figure 3A or the description of the results.

Graphs from Figures 2 and 3 could use the same pattern, especially on the y-axis. Also, you could use a better color pattern for bars in Figure 3, and since Figure 3B uses the same data from the SERPINF1 gene in the 48h group, use the same color from Figure 3A. Make it more presentable, and make it much clearer what you are showing on those graphs.

Regarding the in silico data analysis, did you use any specific database with samples from diabetic subjects? This could bring more strength to the data you show.

At line 122, check for the spelling of RT-qPCR.

Finally, check for the correct way to represent human gene names. You shall write it all uppercase and italics.

6. PLOS authors have the option to publish the peer review history of their article (what does this mean?). If published, this will include your full peer review and any attached files.

Reviewer #1: No

Reviewer #2: No

---

## [Author Response · Author response to Decision Letter 0]

19 Jul 2024

Subject: Submission of revised paper [PONE-D-23-37569]

Submission Date: July 13, 2024

Partha Mukhopadhyay, Ph.D

Section Editor

PLOS ONE

Dear Dr. Mukhopadhyay,

Thank you for inviting us to submit a revised draft of our manuscript entitled, “Crosstalk between miR-96 and IRS/PI3K/AKT/VEGF cascade in hRPE cells; A potential target for preventing diabetic retinopathy” to PLOS ONE journal. We also appreciate the time and effort you and each of the reviewers have dedicated to providing insightful feedback on ways to strengthen our manuscript. Thus, it is with great pleasure that we resubmit our article for further consideration. We have incorporated changes that reflect the detailed suggestions you have graciously provided. We also hope that our edits and the responses we provide below satisfactorily address all the issues and concerns you and the reviewers have noted.

Best regards,

Zahra-Soheila Soheili 

Ph.D. in Biochemistry

Department of Molecular Medicine, National Institute of Genetic Engineering and Biotechnology

P.O.Box: 14965/161, Pajoohesh Boulevard, 17th Kilometers, Tehran-Karaj Highway, 

Tehran-Iran

Tel: +98-21-44787379

Fax: +98-21-44787399

Email: soheili@nigeb.ac.ir

ORCID ID: https://orcid.org/0000-0003-1292-465X

Reviewer 1 Comments:

1. This manuscript is series of genes based on gene and protein interaction networks and utilized text mining resources. To examine the miR-96 impact on candidate genes expression, we overexpressed miR-96 via adeno-associated virus (AAV)-based plasmids in retinal pigment epithelial (RPE) cells. This manuscript is good in shape well described.

I'm happy to accept this manuscript.

 RESPONSE: 

Dear Reviewer,

We are deeply grateful for the time and attention you have devoted to reviewing our submission. We are also honored to have your positive feedback on our manuscript. Thank you again for the time and effort you have invested in reviewing our manuscript.

Respectfully,

Zahra-Soheila Soheili

Ph.D. in biochemistry

Department of Molecular Medicine, National Institute of Genetic Engineering and Biotechnology

Reviewer 2 Comments:

1. The authors started from the premise that microRNA from the family of miR-183 is overexpressed in diabetic retinopathy. They chose miR-96, which is increased in animal models for diabetes to study. The authors then induced miR-96 overexpression in a culture of human retinal pigment epithelium (hRPE) cells through an adenoviral vector. They showed changes in genes related to insulin response and diabetes development, which shows that miR-96 may be important to the development of diabetic retinopathy.

In a wide sense, the paper is well-written, with few grammatical problems (pay attention to some definite articles that don’t seem necessary), yet I recommend it to be double-checked by some specialists.

 RESPONSE: 

Dear Reviewer,

Thank you for taking the time to carefully review our manuscript and provide such thoughtful and detailed feedback. We greatly appreciate the effort and attention you have dedicated to improving the quality of our manuscript. Your comments have been very valuable in helping us identify areas for clarification, strengthen our analysis, and refine the overall presentation. Your input has been invaluable, and we sincerely thank you for contributing your time and expertise to this project.

Our team has thoroughly proofread the manuscript and made the necessary corrections to improve the quality of the writing. We are confident that the manuscript is now free of any major grammatical problems.

2. Grammar aside, I found the text hard to follow. It lacks connectivity and a cause/effect sense of your experiments.

 RESPONSE: We appreciate your feedback on our text. We apologize for any difficulties you experienced in following the manuscript and, understand your concern about the clarity and flow of the text. To enhance the connectivity, we focused on incorporating appropriate transitional phrases and connecting and relating ideas in the manuscript. To address the experiment’s lack of a clear cause/effect sense, we revised sentences to convey the relationships between variables and make it clear to the reader.

3. The bioinformatic segment doesn’t seem to be organically inserted in the text, it seems lost in the context.

 RESPONSE: Thank you for your constructive comment. We agree that this is an important point to address for the clarity and coherence of our paper. To rectify this concern, we reevaluated the placement and current positioning of the bioinformatic segment within the text. Following your suggestion, we have relocated the position of bioinformatic section. We have also revised the manuscript to provide a concise overview of the bioinformatic databases employed, along with relevant references to improve the organic integration of the bioinformatic segment within our paper.

4. The in vitro part doesn’t feel like it is leading to the biosystem analysis.

 RESPONSE: We appreciate your comment and would like to provide some additional context that may help address your concerns about the flow between the in vitro and biosystem analysis. Before performing the in vitro experiments, we did in silico research to identify key target genes which helped inform the selection of specific genes and pathways for further investigation. We also explored additional databases to predict miRNA-pathway interactions and retrieve the association between genes and diabetic retinopathy to present the in vitro findings in the broader biosystem by aligning the data with our experimental results.

By modifying the placement of the in silico section in the manuscript as per your recommendation, we believe the logical flow of the study is now much better defined.

5. As far as I understood, your in silico data would be better presented first, to be later confirmed by the in vitro experiments.

 RESPONSE: We are thankful for your guidance in improving the presentation of our research. We have updated the manuscript by moving the in-silico data to the beginning of the Material and Methods and the Results section (page 6, line 102, and page 11, line 196). We believe this reformed structure will present a better logical progression of our work.

6. This manuscript could use some better storytelling so whoever comes to read your paper might follow it with lesser confusion.

 RESPONSE: Thank you for providing these insights. Scientific writing can sometimes become overly technical and fail to engage the reader, so your suggestion to focus on better storytelling is well taken. To address this, we reorganized certain sections as you mentioned an item in comment 5. We have also added more detailed clarifications throughout the manuscript to ensure that the concepts and findings are clearly communicated. To further enhance the reader’s understanding, we have included a schematic diagram (Figure 1) that illustrates the key steps performed in this study. We hope these modifications have addressed your feedback satisfactorily.

7. For RT-qPCR you used TaqMan probes from Qiagen, right? I would like you to describe the probes you used, including their catalog number, in the Material and Method section.

 RESPONSE: Thank you for your question about the methodology used for the RT-qPCR experiment in our manuscript. We can confirm that we did not utilize TaqMan probes for this part of our study. Instead, we used the dye-based method using SYBR Green as the intercalating fluorescent dye.

As described in the Material and Method section of the manuscript (page 8, lines 161 - 169), reaction conditions and other technical details of the SYBR Green-based RT-qPCR approach were provided. We felt this method was appropriate for the goals of our study and provided a cost-effective, sensitive way to quantify gene expression.

Also, we have added the catalog number of QuantiFast SYBR® Green PCR Kit (Qiagen, Germany) to the Material and Method section (page 8, line 167).

8. Regarding your PCR results, it is not clear how you analyzed them. You have basically two groups, control and the one treated with the miR-96 overexpressing AAV. For 2ΔΔCt analysis, you need at least two groups, where one of those would be the control and the other(s) would be times-fold its expression. How did you analyze your data, though? Did you test all those genes for both groups and show only the treated group of each gene in Figure 3A? If so, I don’t have much problem with this way of representing the results, since you keep it very clear in the text, and in the figure legend what you did.

 RESPONSE: Thank you for providing these insights. As you mentioned, we had two groups for 2ΔΔCt analysis. The control group (normal hRPE cells) and treated group (hRPE cells in which MIR96 was upregulated by more than 82-fold). We tested all genes for both groups, and as you pointed, we reported the relative expression of each gene in the treated group compared to the control group in Figure 7A.

We have rewritten the description of this analysis to be more in line with your comment (page 9, lines 171-173).

9. For Figure 3B, what was the statistical analysis you used?

 RESPONSE: Thank you for your inquiry regarding the statistical analysis employed in Figure 3B (now renamed to Figures 7B due to reorganization of manuscript). We used two-way repeated measures ANOVA as a suitable statistical analysis method considering our data on gene expression at different time points. This analysis allowed us to compare the expression of FOXO1 and SERPINF1 genes across the 24-hour and 48-hour time points.

10. Nonetheless, about the statistics, was there any normality test used? It is important to determine if the data was properly analyzed by the tests you described since they are only for normal-distributed data.

 RESPONSE: You have raised an important point considering the data distribution and its impact on statistical analysis. 

All real-time PCR reactions were validated according to the MIQE guidelines. We confirmed that our data followed a normal distribution using the Shapiro-Wilk test, which verified that our variability measures were normally distributed. These rigorous checks ensured that our gene expression results were precise and reproducible, enhancing the robustness of our findings, which were included in both published and unpublished works.

11. Table 1 description (line 220) is weird. A P-value equal to or below 0.05 is already statistically significant, for what you described in the methods. As a matter of fact, Table 1 seems redundant. If you want to highlight the p-values, just add them in Figure 3A or the description of the results.

 RESPONSE: Thank you for your suggestion regarding Table 1 in the manuscript. We agree with you and decided to remove Table 1 entirely. Instead, we have incorporated p-values into Figure 6 description to present information in a single Figure and avoid unnecessary data (page 15, line 279).

12. Graphs from Figures 2 and 3 could use the same pattern, especially on the y-axis. Also, you could use a better color pattern for bars in Figure 3, and since Figure 3B uses the same data from the SERPINF1 gene in the 48h group, use the same color from Figure 3A. Make it more presentable, and make it much clearer what you are showing on those graphs.

 RESPONSE: We appreciate your suggestions to improve our data visualization. We agree that these elements could be optimized for a better presentation.

We have updated the graphs in Figures 2 and 3 (now renamed to Figures 6 and 7 due to the reorganization of the manuscript) to address your concerns. We have tried to make the y-axis of the graphs as similar as possible. We have also altered the color scheme and selected an easily distinguishable palette, especially for SERPINF1 and FOXO1 genes. The updated graphs now offer a clearer look and we hope they align with your recommendations.

13. Regarding the in silico data analysis, did you use any specific database with samples from diabetic subjects? This could bring more strength to the data you show.

 RESPONSE: We appreciate your insightful feedback. We have utilized the DisGeNet database as the source of diabetic retinopathy subject samples (page 6, line 115). DisGeNet is a comprehensive platform that integrates information on gene-disease associations from several sources, including scientific literature, expert-curated databases, and genome-wide association studies. This robust dataset enabled us to conduct our analysis with a high degree of confidence and to draw meaningful conclusions regarding the genetic factors related to this important diabetic complication.

14. At line 122, check for the spelling of RT-qPCR.

 RESPONSE: We apologize for this error. We’ve corrected the typo [RT-Qpcr] to [RT-qPCR] (page 8, line 157).

15. Finally, check for the correct way to represent human gene names. You shall write it all uppercase and italics.

 RESPONSE: Thank you for pointing this out. The gene names have all been corrected to the format that you mentioned. They are highlighted in yellow throughout the manuscript.

Again, thank you for giving us the opportunity to strengthen our manuscript with your valuable comments and queries. We have worked hard to incorporate your feedback and hope that these revisions persuade you to accept our submission.

Best regards,

Zahra-Soheila Soheili 

Ph.D. in Biochemistry

Department of Molecular Medicine, National Institute of Genetic Engineering and Biotechnology

P.O.Box: 14965/161, Pajoohesh Boulevard, 17th Kilometers, Tehran-Karaj Highway, 

Tehran-Iran

Tel: +98-21-44787379

Fax: +98-21-44787399

Email: soheili@nigeb.ac.ir

ORCID ID: https://orcid.org/0000-0003-1292-465X

---

## [Decision Letter · Decision Letter 1]

11 Sep 2024

Crosstalk between miR-96 and IRS/PI3K/AKT/VEGF cascade in hRPE cells; A potential target for preventing diabetic retinopathy

PONE-D-23-37569R1

Dear Dr. Soheili,

We’re pleased to inform you that your manuscript has been judged scientifically suitable for publication and will be formally accepted for publication once it meets all outstanding technical requirements.

Kind regards,

Partha Mukhopadhyay, Ph.D.

Section Editor

PLOS ONE

Additional Editor Comments (optional):

Reviewers' comments:

Reviewer's Responses to Questions

**Comments to the Author**

1. If the authors have adequately addressed your comments raised in a previous round of review and you feel that this manuscript is now acceptable for publication, you may indicate that here to bypass the “Comments to the Author” section, enter your conflict of interest statement in the “Confidential to Editor” section, and submit your "Accept" recommendation.

Reviewer #1: All comments have been addressed

Reviewer #2: All comments have been addressed

2. Is the manuscript technically sound, and do the data support the conclusions?

Reviewer #1: Yes

Reviewer #2: Yes

3. Has the statistical analysis been performed appropriately and rigorously? 

Reviewer #1: Yes

Reviewer #2: Yes

4. Have the authors made all data underlying the findings in their manuscript fully available?

Reviewer #1: Yes

Reviewer #2: Yes

5. Is the manuscript presented in an intelligible fashion and written in standard English?

Reviewer #1: Yes

Reviewer #2: Yes

6. Review Comments to the Author

Reviewer #1: Authors are addressed and proper responses of my comments. I'm happy to accept current form of manuscript.

Reviewer #2: I appreciate all the efforts you have taken on answering my observations, and I understand the manuscript has improved and is proper for publication. Congratulations to the authors.

7. PLOS authors have the option to publish the peer review history of their article (what does this mean?). If published, this will include your full peer review and any attached files.

Reviewer #1: No

Reviewer #2: No

---

## [Editor Report · Acceptance letter]

20 Sep 2024

PONE-D-23-37569R1 

PLOS ONE

Dear Dr. Soheili, 

I'm pleased to inform you that your manuscript has been deemed suitable for publication in PLOS ONE. Congratulations! Your manuscript is now being handed over to our production team.

Kind regards, 

on behalf of

Dr. Partha Mukhopadhyay 

Section Editor

PLOS ONE